



# Last glacial millennial-scale hydro-climate and temperature changes in Puerto Rico constrained by speleothem fluid inclusion δ¹⁸O and δ²H values

Sophie F. Warken[1,2*], Therese Weissbach[1,3*], Tobias Kluge[1,3,4], Hubert Vonhof[5], Denis Scholz[6], Rolf Vieten[7], Martina Schmidt[1], Amos Winter[7,8], and Norbert Frank[1]

[1] Institute of Environmental Physics, University of Heidelberg, Heidelberg, Germany
[2] Institute of Earth Sciences, University of Heidelberg, Heidelberg, Germany
[3] Heidelberg Graduate School of Fundamental Physics, Heidelberg University, Heidelberg, Germany
[4] Institute of Applied Geosciences, Karlsruhe Institute of Technology, Karlsruhe, Germany
[5] Max Planck Institute for Chemistry, Climate Geochemistry Department, Mainz, Germany
[6] Institute for Geosciences, University of Mainz, Germany
[7] Department of Marine Sciences, University of Puerto Rico, Mayagüez, Puerto Rico
[8] Earth and Environmental Systems Department, Indiana State University, Terre Haute, Indiana, USA

*Both authors contributed equally to this work

*Correspondence to*: Sophie Warken (swarken@iup.uni-heidelberg.de)

## Abstract

We present speleothem fluid inclusion $\delta^{18}O_f$ and $\delta^2H_f$ values from Larga Cave, Puerto Rico, that covers the interval between 46.2 to 15.3 ka before present on millennial scale, including the Last Glacial Maximum (LGM) and several stadial and interstadial cycles. The dataset can be divided in two main clusters of stable isotope compositions of the fluid inclusion water with respect to the global meteoric water line which coincide with strong variations in the water content of the stalagmite. In particular, this clustering is found to be climate related, where the first cluster comprises samples from cold and dry periods, such as Heinrich and Greenland stadials, as well as parts of the LGM, which exhibit very high $\delta^{18}O_f$ and $\delta^2H_f$ values. We interpret this enrichment as caused by evaporation inside the cave due to enhanced cave ventilation during these colder and drier times. In contrast, in most samples corresponding to warmer and wetter Greenland interstadials, but also for some from Heinrich Stadial 2 and 3, the $\delta^{18}O_f$ and $\delta^2H_f$ values plot on the meteoric water line and modification of fluid inclusion water due to "in-cave" evaporation is found negligible.

Consequently, variations of last glacial hydro-climate and temperature in the western tropical Atlantic can be constrained. In general, $\delta^{18}O_f$ values from fluid inclusions are up to 3‰ higher than those of modern drip water, which is interpreted as a weaker atmospheric convective activity during the last glacial period. In addition, reconstructed temperatures suggest an average cooling of c. 3°C during the LGM compared to modern cave temperature. During Heinrich Stadials 2 and 3, reconstructed cave temperatures yield an additional cooling of 2.9 ± 2.6°C and 4.4 ± 0.6°C, respectively. Higher $\delta^{18}O_f$ values of these samples further suggest that the drip water was dominated by orographic rainfall and/or cold fronts, along with weak





or even absent convective activity. In contrast, during interstadial phases, reconstructed temperatures reached nearly modern

values, and convective activity was comparable or only slightly weaker than today.

## 1   Introduction

Stable isotopes of oxygen (represented by $\delta^{18}O$ values) are a widely used proxy to deduce information about past climates from speleothems, since the cave drip water reflects the amount- or recharge-weighted annual mean isotopic signature of the rainfall above the cave (Baker et al., 2019;Lachniet, 2009). This value is in turn modulated by several variables, such as

temperature, the type and seasonal distribution of precipitation as well as the source moisture trajectory (Lachniet, 2009;Bird et al., 2020;Lases-Hernandez et al., 2020). In the tropics, the isotopic composition of precipitation, $\delta^{18}O_p$, has been attributed to changes in precipitation amount and convective activity (Dansgaard, 1964;Lachniet and Patterson, 2009;Vieten et al., 2018a). Consequently, tropical speleothem $\delta^{18}O$ values are usually interpreted in terms of this isotopic 'amount effect' (Medina-Elizalde et al., 2010;Arienzo et al., 2019;Lases-Hernandez et al., 2019). However, in the soil and karst above the cave

as well as at the surface of a stalagmite, various (disequilibrium) isotope fractionation effects can obscure the interpretation of stable isotopes in speleothem calcite (Weber et al., 2021;Hansen et al., 2019;Carlson et al., 2020).

In this respect, stable isotopes of oxygen and hydrogen in speleothem fluid inclusions provide a useful proxy, since these represent potentially unchanged aliquots of the original drip water, and thus relicts of past precipitation (Schwarcz et al., 1976;van Breukelen et al., 2008;Affolter et al., 2015). The utility of speleothem fluid inclusion analysis improves if the

conditions of carbonate precipitation are well understood, i.e., whether or not the minerals precipitated close to 'equilibrium conditions'. In the case of equilibrium conditions, paleo-temperatures can be reconstructed from combined oxygen isotope ratios of the calcite ($\mathbf{\delta^{18}O_c}$) and the fluid inclusions, $\mathbf{\delta^{18}O_f}$, based on known temperature dependent isotope fractionation relationships between calcite and water (e.g., Kim and O'Neil (1997);Tremaine et al. (2011);Hansen et al. (2019);Johnston et al. (2013). In continental mid- and high latitude locations, estimations of paleo-temperatures via fluid inclusion $\delta^{2}H_f$ values

alone are feasible, based on the dominant control of temperature on rainfall isotope composition in such settings (Affolter et al., 2019;Demény et al., 2021). This only works, however, if the contribution of the amount effect is negligible. The analysis of speleothem fluid inclusions has been successfully applied worldwide, including tropical locations (Griffiths et al., 2010;Arienzo et al., 2015;Millo et al., 2017;Meckler et al., 2015). Nevertheless, speleothem fluid inclusion records from the Last Glacial Period (c. 115 – 11.7 ka) are still sparse, and in many cases, reliable paleo-temperature estimates are challenging

(Millo et al., 2017;Demény et al., 2021;Nehme et al., 2020).

Here we present a detailed study of fluid inclusions from a Puerto Rican stalagmite (PR-LA-1), spanning the interval between 46.2 to 15.3 ka. The record encompasses a period which was characterized by rapid climatic fluctuations from cold (stadial) to warmer (interstadial) conditions recurring on millennial time scales (Dansgaard et al., 1984;Peterson et al., 2000). In paleoclimate records from the tropics, the so-called 'Heinrich Stadials' (HS) are clearly represented by dramatic cooling and

dry conditions in the western tropical Atlantic (Escobar et al., 2012;Arienzo et al., 2015;Grauel et al., 2016). These conditions





were likely caused by a southward shift of the Inter-Tropical Convergence Zone (ITCZ) induced by a near shutdown of the Atlantic Meridional Overturning Circulation (AMOC) (Broccoli et al., 2006;Lynch-Stieglitz et al., 2014;Deplazes et al., 2013;Escobar et al., 2012). Conversely, during the interstadials, the ITCZ was presumably in a more northerly position, accompanied by warmer and wetter climatic conditions and higher SSTs in the (sub-) tropical Atlantic (Schmidt et al.,
2006a;Peterson et al., 2000;Sachs and Lehman, 1999).

The speleothems PR-LA-1 was selected because its stable isotope values ($\delta^{18}O_c$ and $\delta^{13}C_c$) and Mg/Ca ratios demonstrated a pronounced response to local hydro-climate on the millennial-scale (Warken et al., 2020). To further elucidate the millennial-scale climate changes during the last glacial stadials and interstadials, we thus conducted a large number of fluid inclusion measurements following prominent $\delta^{18}O_c$ changes in stalagmite PR-LA-1.


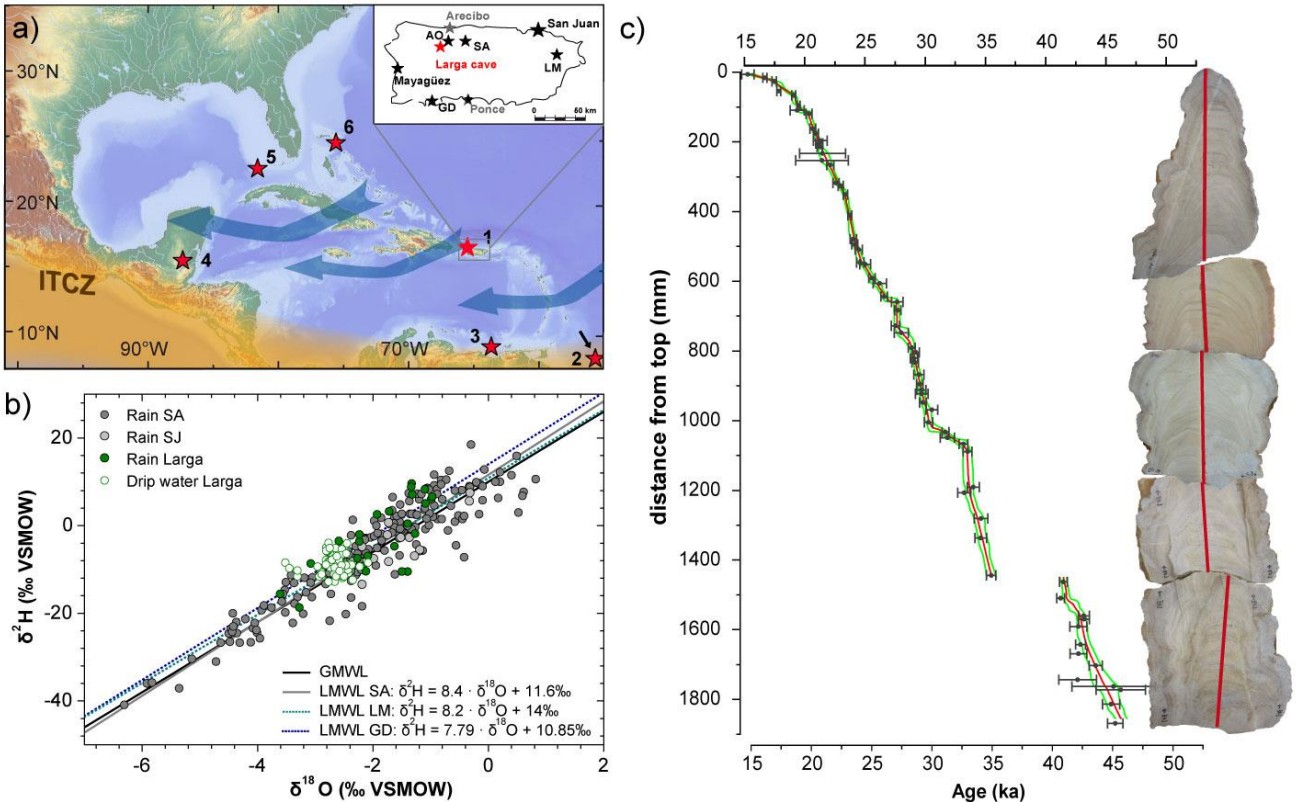

**Figure 1: Sample locations and previous work on stalagmite PR-LA-1. a)** Location of Larga Cave (1), Puerto Rico, in the western tropical Atlantic region and other sites discussed in this study (map after Warken et al. (2020)) : (2) Guiana Basin (Rama-Corredor et al., 2015); (3) Cariaco Basin, Venezuela (Deplazes et al., 2013); (4) Lake Petén Itzá, Guatemala (Escobar et al., 2012;Grauel et al.,
2016); (5) Florida Strait (Them Il et al., 2015); (6) Abaco, Bahamas (Arienzo et al., 2015). The insert shows a detailed view on the location of Larga cave (red star), and the position of local meteorological stations: Arecibo observatory (AO), San Agustín (SA), Luquillo Mountains (LM), and Guánica Dry Forest (GD). Blue arrows in panel A indicate the dominant moisture trajectories over the region in boreal summer, when the Intertropical convergence zone (ITCZ, yellow shaded area) is in its northernmost position. **b)** Local rain water isotope data from Larga Cave (Vieten et al., 2018b), SA (Scholl et al., 2014) and GNIP station San Juan (SJ). In
addition, the GMWL and LMWLs are shown from SA (Putman et al., 2019;Scholl et al., 2014); LM (Scholl and Murphy, 2014) and GD (Govender et al., 2013). **c)** Image of stalagmite PR-LA-1 and age-depth-model from Warken et al. (2020).





## 2    Material and Methods

### 2.1    Site description

Larga Cave (18°19'N 66°48'W, Fig. 1) is located 350 m above sea level in the north central karst region of Puerto Rico. The
overburden of Larga Cave is comprised of the Oligocene Lares limestone Fm., which is covered by a thin layer of soil. The
host rock is exposed at some places, and vegetation consists today of dense tropical forest. The main passage in the cave is
almost horizontal and has a total length of 1400 m with a ceiling height of up to 30 m (Miller, 2010). Stalagmite PR-LA-1 has
a total length of 185 cm and was found lying on the cave floor in the main passage, where the rock overburden is 40 to 80 m
thick. The stalagmite was deposited between 15.4 to 46.2 ka before present with a hiatus between 35.5 and 41.1 ka (Fig. 1,
Warken et al. (2020)). Between 2012 and 2017, the mean annual air temperature at the location of the cave was 22.5°C, and
measured local annual rainfall amount had a mean of 2,200 mm (Vieten et al., 2018a). Since 2012, cave air parameters such
as temperature (T), $pCO_2$ and relative humidity (RH), have been recorded at the position of stalagmite PR-LA-1. During this
time, the cave temperature was constant with values of 22.5±0.2 °C. The cave air $pCO_2$ varies depending on the season, with
high values of up to 1800 ppm in summer and low values of around 550 ppm in winter (Vieten et al., 2016). This seasonal
pattern results from a temperature-driven, very well-ventilated cave environment in winter and low ventilation in summer.
During the winter season, a high buoyancy contrast at night leads to maximal cave ventilation, while it is at a minimum during
summer, when cave temperatures are lower than in the outside environment. RH is mostly close to 100%, but has been found
to decrease to values to 90-95% for several weeks when ventilation intensifies in winter months (Vieten et al., 2016;2018a).

As typical for Caribbean locations, the precipitation regime over Larga Cave is characterized by a drier winter season, from
December to April and a much wetter season during the rest of the year. During the drier season, occasional light showers
occur imbedded in trade winds, which are interrupted by occasional cold fronts emanating from North America (Scholl and
Murphy, 2014). In contrast, during the remainder of the year convective rain is formed by easterly moving low pressure systems
which occasionally develop into tropical depressions accounting for the major part of total precipitation (Scholl and Murphy,
2014). On inter-annual time scales, the $\delta^{18}O$ values of rainfall in Puerto Rico show a negative correlation with precipitation
suggesting an isotopic 'amount effect' of c. -0.1 ‰ per 100 mm of rainfall amount (Govender et al., 2013;Scholl et al.,
2014;Warken et al., 2020). Over the course of monthly drip water monitoring (2013 – 2017), the drip water composition at
sites with a maximum distance of 10m to the location where stalagmite PR-LA-1 was found shows only little seasonal variation
with a mean value of $\delta^{18}O_w = -2.6 \pm 0.3‰$, and $\delta^2H_w = -9.5 \pm 0.2‰$ (VSMOW). The seasonal signal in the rain water is
strongly smoothed by the soil and epikarst above the cave with a mixing time of at least several months to years, suggesting
that Larga speleothems time resolution record is at least inter-annual.

A local meteoric water line (LMWL) is available for three stations in Puerto Rico (Fig. 1). At San Agustin, located closest to
Larga cave, a monthly dataset covering 1998 to 2012 shows a LMWL of $\mathbf{\delta^2H = 8.4 \times \delta^{18}O + 11.6‰}$ (VSMOW) (Putman et



al., 2019;Scholl et al., 2014). Similar LMWLs (Fig. 1) exist for a site in the Luquillo Mountains in eastern Puerto Rico (Scholl and Murphy, 2014) as well at Guánica Dry Forest located in south-western Puerto Rico (Govender et al., 2013).

## 2.2    Fluid inclusion stable isotope composition

Solid samples for fluid inclusion isotope analysis were cut using a band saw along the growth axis of stalagmite PR-LA-1. High growth rates of this specimen of up to 1mm per year allow the sampling of distinct millennial scale climate oscillations evident in the $\delta^{18}O_c$ record, such as D/O or Heinrich events. Replicate samples were taken along the corresponding growth layers to minimize the effect of integrating over too long time intervals. Individual sample sizes were between 0.21 – 1.34 g.

The speleothem fragments were placed in a copper tube and installed in a preparation line connected to cavity ring down spectrometer (CRDS, L2130-I, Picarro) at the Institute of Environmental Physics, Heidelberg University (Weißbach, 2020). The samples were subsequently crushed using a hydraulic press. To avoid the CRDS memory effect, a stable water vapour background was generated to ensure constant saturation of the cavity without condensation (Affolter et al., 2014;De Graaf et al., 2020). This was achieved by mixing dry nitrogen as a carrier gas with water of known isotopic composition, in a line held

at a constant temperature of 120°C.

The mixture of background water and sample signal is measured in the CRDS analyser, and evaluated similar to the method of Affolter et al. (2014) with a Python code. The isotopic composition and released water volume is calibrated with in-house water standards and glass capillaries with known volumes in the µl range, respectively. These are inserted into the copper tube and connected to the preparation line, similarly as the speleothem samples. The in-house water standards are analysed regularly

against the international primary reference material VSMOW2 and SLAP2. The long-term precision of single stable isotope measurements is 0.5 ‰ for $\delta^{18}O$ and 1.5 ‰ for $\delta^2H$, while for released water volumes above 1.0 µl, the precision improves to 0.2‰ for $\delta^{18}O$ and 0.4‰ for $\delta^2H$ (Weißbach, 2020). In case of water volumes < 0.2 µl, the result of a single measurement may become less precise and is therefore reproduced by replicate measurements to verify the result. If not stated differently, the stable isotope composition of fluid inclusions is given relative to Vienna Standard Mean Ocean Water (VSMOW).




**Figure 2: Results of fluid inclusion analysis of stalagmite PR-LA-1 (Table S2). From top to bottom: (a) Growth rate (logarithmic scale) and (b) δ18Oc values from Warken et al. (2020). Symbols indicate the averaged δ18Oc values used for each depth of fluid inclusion isotope analysis, respectively. Vertical blue bars (dotted red lines) indicate the timing of cool/dry (warm/humid) periods, such as Heinrich stadials HS1 to HS3 (Greenland interstadials GI12c to 6) in the western tropical Atlantic (Warken et al., 2020). The following panels show fluid inclusion isotope values of (c) δ18Of, (d) δ2Hf, and (e) measured water content. Symbol colors indicate the samples with high (dark colored symbols) and low (light colored stars) water content with a threshold value of 0.55 µl/g, respectively. Horizontal lines indicate modern cave conditions. Note that the axes of the δ18Oc, δ18Of, and δ2Hf values are inversed.**





## 3    Results

### 3.1    Water content and stable isotope composition of speleothem fluid inclusions

In total, 64 fluid inclusion water isotope measurements were performed from 30 different depth intervals. The results of the single measurements are shown in Table S1, while the mean values of the replicate measurements are given in Table S2. The mean water content of PR-LA-1 calcite for the different depth intervals shows a variation from close to 0 to 3 µl/g (Fig. 2). Sections with a relatively high water content, e.g., around c. 28 or 34ka also show a high (or at least highly variable) growth rate, while parts with a comparably low water content, such as at c. 20, 26, or 32ka are mainly in sections with relatively slow growth rates. The mean fluid inclusion water $\delta^{18}O_f$ values vary between -1.9 and +11.4‰, and mean $\delta^2H_f$ values are between +6.9 and +44.9‰ (VSMOW) (Fig. 3a). The data exhibit a systematic $^{18}O$ and $^2H$ enrichment compared to modern drip water stable isotope values. This general pattern is also reflected in the slope of the linear regression line fitted to the data, which is with 2.42 ± 0.25 significantly lower than expected for a LMWL. This slope agrees with those of globally distributed evaporation lines for lakes and soil waters (Gibson et al., 2008) as well as evaporation experiments (Hu et al., 2009). Furthermore, the data can be allocated to two different clusters, depending on the measured water content of the samples. The first cluster comprises samples which are mainly located on or scatter around the GMWL. Most of these are also characterized by a water content greater than 0.55 µl/g. In contrast, a lot of samples show more positive $\delta^{18}O$ and $\delta^2H$ values and clearly plot off the GMWL along the linear regression line. In contrast to the first set of samples, most of these have a water content lower than 0.55 µl/g. Only one sample clearly does not fit into this pattern and plots away from the GMWL despite a water content >0.55µl/g (sample 32b), which therefore are also allocated to cluster (2). Conversely, two samples plot close to the MWLs despite a relatively low water content (samples 5 and 8).

### 3.2    Calculation of paleo-temperatures using the oxygen isotope equilibrium thermometer

In order to assess the applicability of the oxygen isotope equilibrium thermometer based on the mean fluid inclusion $\delta^{18}O_f$ values, the average carbonate $\delta^{18}O_c$ values were calculated from the intervals covered by the depth range of the measured fluid inclusion samples, respectively (Table S2). Figure 4a shows that the previously mentioned pattern with two clusters of samples is also reproduced when comparing the fluid inclusion $\delta^{18}O_f$ with the carbonate $\delta^{18}O_c$ values. Samples with a high water content, which plot on the GMWL in Fig. 3a, are clearly separated by the samples with a lower water content. While cluster (1) samples with a high water content show a positive correlation between $\delta^{18}O_c$ and $\delta^{18}O_f$ values, cluster (2) samples do not follow the same trend. Similarly, the fractionation between speleothem calcite and fluid inclusions, $^{18}\varepsilon_{c/f} = 1000 \cdot \ln\alpha$ (with $\alpha = (1000 + \delta^{18}O_c,)/(1000 + \delta^{18}O_f)$, and $\delta^{18}O_c$ and $\delta^{18}O_f$ in VSMOW) shows a clear dependence on the $\delta^{18}O_f$ value for the evaporated samples of cluster (2) (Fig. 4b), but no relationship with carbonate $\delta^{18}O_c$ values (Fig. 4c). In contrast, for the cluster (1) samples, no relationship between $^{18}\varepsilon_{c/f}$ and $\delta^{18}O_f$ values is observed (Fig. 4b). Figure 4 also suggests that fluid inclusion samples 9b, 32b, 27, 32, and 36 are also enriched in $^{18}O$, and have to be allocated to cluster (2).



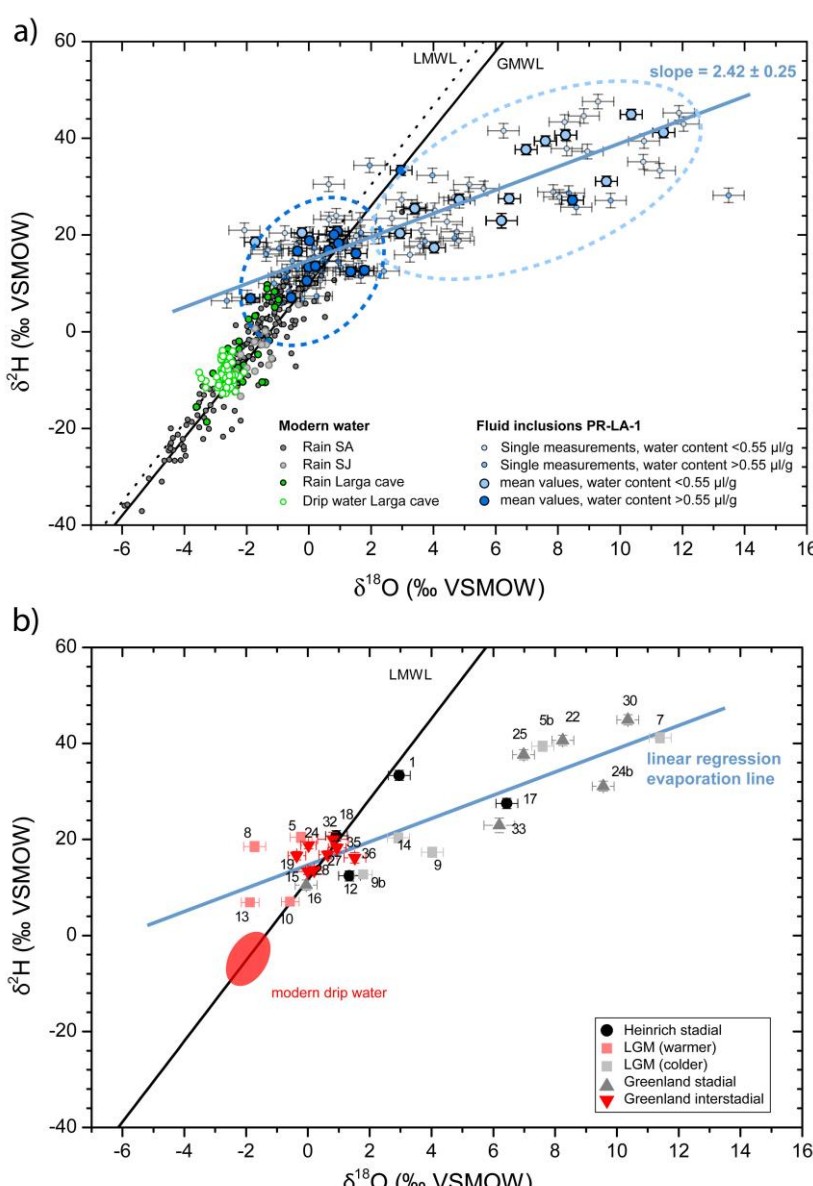

**Figure 3: a) Results of stable isotopes of fluid inclusions and drip waters in Larga Cave as indicated in Tables S1 and S2. Lines indicate the GMWL as well as the LMWL with δ²H = 8.2· δ¹⁸O+14‰ VSMOW by Scholl and Murphy (2014). Single values of the water isotopes in fluid inclusions (small circles) and mean values (large circles) with corresponding standard deviation are shown together with Larga cave drip water from different drip sites (2013 - 2019, green triangles) and local rain water data (compare Fig. 1). The mean values of the replicate measurements are allocated to two groups, with cluster (1) comprising samples on or scattering around the GMWL (indicated by the dark blue circle), and cluster (2) comprising all samples with a clear evaporation signal, plotting off the GMWL (light blue circle). b) Stable isotope values of fluid inclusions from stalagmite PR-LA-1, categorized according to their allocation to the presumably warm (interstadial) or cold (stadial) periods when they formed. Thereby, a distinction is made between Heinrich stadials, Greenland stadials, and Greenland interstadials. Samples analysed from the LGM are also subdivided in presumably more warm/humid and relatively cool/dry periods. Numbers refer to the sample IDs as shown in Tables S1 and S2.**





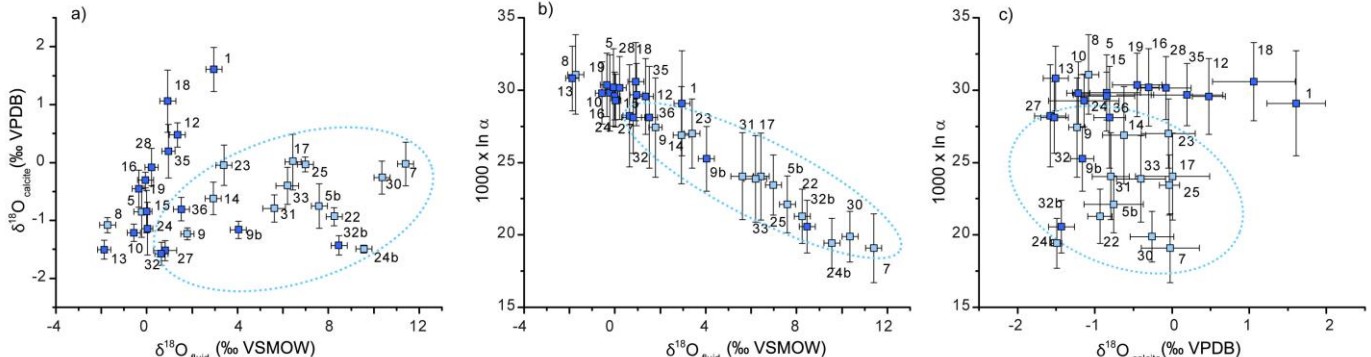

**Figure 4: Analysis of the fractionation between speleothem calcite and fluid inclusions. Panel (a) shows the measured δ¹⁸Oc values in speleothem carbonate (Warken et al., 2020) vs. measured δ¹⁸Of in speleothem fluid inclusion water (this study). (b) fractionation δ c/f = 1000 lnα between carbonate and fluid inclusion water vs. the δ¹⁸Of values. (c) fractionation ¹⁸εc/f = 1000·lnα vs. δ¹⁸Oc values. Numbers indicate the sample IDs of the fluid inclusion measurements (Table S2). The uncertainty of the calculated fractionation is estimated from propagating the analytical error of the fluid inclusion measurements, as well as the range of the δ¹⁸Oc calcite value due to dating and sampling uncertainty. Blue symbols indicate samples with a water content >0.55 µl/g, and light blue symbols a water content <0.55 µl/g. The ellipses indicate the samples which are allocated to cluster (2) due to large fractionation effects in δ¹⁸Of.**

For a first-order temperature calculation from $^{18}\varepsilon_{c/f}$, three different isotope fractionation relationships have been tested (Kim and O'Neil, 1997;Tremaine et al., 2011;Johnston et al., 2013). All results are listed in Table S2, where the given uncertainties of the paleo-temperatures result from propagating the analytical error of the fluid inclusion measurements, as well as the range of the $\delta^{18}O_c$ calcite value due to dating and sampling uncertainty. All paleo-temperatures reported and discussed in the main text are calculated using the Johnston et al. (2013) equation, which has been developed from cave monitoring and modern speleothem calcite and yields realistic temperatures compared to the modern cave air temperature (Table S2). Note that we discard samples from the temperature discussion where replicate measurements exhibit a poor reproducibility in terms of largely variable water content and large isotopic differences, in order to focus on the most robust paleo-climatic inferences (Table S2). Note also that we did not calculate temperatures for the samples of cluster (2), which did not contain sufficient water and clearly show significant influence from fractionation through evaporation (Table S2). We regard the temperature results obtained on the 12 replicate samples from cluster (1) as reliable within the large uncertainty of 1.5 – 2.5°C. The calculated temperatures scatter around the modern cave temperature of 22.8°C, with paleo-temperatures varying between 15.2 ± 1.7 °C (sample 8, 21.39 ± 0.25 ka) and 24.0 ± 1.2 °C (sample 24, 33.52 ± 0.52 ka), and a mean value of 20.0 ± 2.5 °C.



## 4    Discussion

### 4.1    Influence of evaporation on PR-LA-1 fluid inclusion isotope data

#### 4.1.1    Categorization of the fluid inclusion isotope data

For the fluid inclusion isotope data from cluster (1), $\delta^{18}O_f$ and $\delta^2H_f$ values scatter around the LMWL (Govender et al., 2013;Putman et al., 2019;Scholl and Murphy, 2014;Scholl et al., 2014). Thus, we suggest that these samples represent pristine fossil drip water. The second cluster of fluid inclusion isotope values contains mainly samples with water contents below 0.55µl/g, which all show increased $\delta^{18}O_f$ values which are noticeably more fractionated than hydrogen isotopes. In addition to the categorization with respect to water content, the fluid inclusion samples can be further associated with stadial and interstadial phases during the Last Glacial according to their stable isotope and elemental signatures in speleothem calcite (Fig. 3b). The fluid inclusion samples representing warm Greenland Interstadial (GI, red triangle) plot close to the MWL and can be all associated with cluster (1). In contrast, the samples formed during the colder intervals (HS; GS, LGM) are mainly attributed to cluster (2) and trend away from the MWL. In addition, for Last Glacial Maximum (LGM) samples, the differentiation between the two clusters appears to be valid, with relatively warm/humid phases plotting closer to the MWL, and relatively dry/cool phases rather clustering with the GS and HS samples (Fig. 3b). This demonstrates that our dataset allows us to distinguish cooler and/or drier climatic background states during Heinrich and Greenland Stadials, represented by cluster (2) samples, from the presumably warmer and more humid phases during Greenland Interstadials, which comprise cluster (1). Note that evaporation driven fractionation must indicate a general annual decrease of humidity in the cave spanning over decades required to trap the water in the calcite. Therefore, we suggest that samples of cluster (2) reflect periods with stronger cave ventilation as recorded today in the thermal contrast of cave and ambient air in winter. This categorisation suggests that the underlying process, which causes this clustering and affects the fluid inclusion isotopic signatures, is purely climate related.

Previous studies have invoked recrystallization-induced oxygen isotope changes in inclusion-hosted water of speleothems (Demény et al., 2016).  This process is unlikely to be systematically related with climate and would shift the fluid inclusion $\delta^{18}O$ values towards more negative values, which is opposite to the observed pattern in this study. Moreover, no systematic signs of diagenetic alteration have been observed for stalagmite PR-LA-1 (Warken et al., 2020). Other studies suggest that *in situ* oxygen isotope exchange of fluid inclusion water with the host carbonate is unlikely to cause significant isotope fractionation (Uemura et al., 2020;Demény et al., 2021). The systematic enrichment of cluster (2) samples indicates that the stable isotope composition of the low water content samples has been affected by fractionation processes in the gas phase as $^{18}O$ is enriched over H isotopes, just as in non-equilibrium evaporation of thin water films (Gat et al., 1994). Evaporation is thus a likely scenario causing the lower slope of the water isotope relationship of $2.42 \pm 0.25$ (Fig. 3) via a temperature and humidity related climate – change (Gibson et al., 2008;Hu et al., 2009).



### 4.1.2 Climate-related evaporation effects on fluid inclusion isotope values in Larga cave

Recent studies have suggested that in-cave evaporation may affect the isotopic composition of speleothem fluid inclusions (Nehme et al., 2020;Weißbach, 2020). A potential climate-related driver of this evaporative fractionation that has not been described previously is temperature-driven ventilation. This effect has been observed in Larga Cave today, especially during the dry winter season (Vieten et al., 2016). Cave air ventilation may provoke evaporation of remaining water at the stalagmite's surface, and thus also from the inclusions that are not yet completely closed, which in turn influences the $\delta^{18}O_f$ and $\delta^2H_f$ values of the inclusion water seconds (Dreybrodt and Scholz, 2011;Deininger et al., 2012;Day and Henderson, 2011). At the same time, evaporation would also reduce the water content of the inclusions, explaining the systematically lower water content of the cluster (2) samples. Since, the speleothems are covered only with a thin film of water, evaporation can be very effective given the high average in cave temperature and assuming a persistently lowered humidity.

Figure 4 shows that the fractionation $^{18}\varepsilon_{c/f}$ between speleothem carbonate and fluid inclusion water is dominated by the enrichment of the fluid inclusion water, which means that the process resulting in the large fractionation of the $\delta^{18}O_f$ values does only have a minor effect on the $\delta^{18}O_c$ values of $CaCO_3$. When regarding both cluster (1) and cluster (2), $^{18}\varepsilon_{c/f}$ is largely independent from $\delta^{18}O_c$ values (Fig. 4c), while a very strong correlation with $\delta^{18}O_f$ values is evident (Fig. 4b). This argues for a significant influence of evaporation, which would have a strong effect on the fluid inclusion water, but only a minor effect on the precipitated carbonate. The reason for the different effect of evaporation on the $\delta^{18}O_f$ values of the fluid inclusion water and the precipitated calcite is related to the very different time-scales of precipitation and oxygen isotope exchange between water and dissolved bicarbonate. Precipitation of calcite occurs on a time scale of a few hundred seconds (Dreybrodt and Scholz, 2011), depending – amongst other parameters - on cave temperature and film thickness. The time-scale of oxygen isotope exchange between water and dissolved bicarbonate, in contrast, is in the range of hours to days (Beck et al., 2005;Dreybrodt and Scholz, 2011), thus orders of magnitude longer.

When the drip water enters the cave, it is reasonable to assume that the $\delta^{18}O$ value of the dissolved bicarbonate is in oxygen isotope equilibrium with the $\delta^{18}O$ value of the drip water. When the drip water then falls onto the surface of a stalagmite, degassing of excess $CO_2$ occurs within a few seconds (Dreybrodt and Scholz, 2011) with a negligible effect on the $\delta^{18}O$ value of the dissolved bicarbonate. Then, precipitation of $CaCO_3$ starts immediately, and the $\delta^{18}O$ value of the bicarbonate that was previously equilibrated with the drip water is imprinted on the $\delta^{18}O$ value of the precipitated calcite. We note that oxygen isotope fractionation during precipitation of speleothem calcite is a complex process (e.g., Dreybrodt and Scholz, 2011;Guo and Zhou, 2019;Hansen et al., 2019;Scholz et al., 2009), and oxygen isotope fractionation does in most cases not occur under conditions of oxygen isotope equilibrium. During precipitation of speleothem calcite, the drip water remaining on the surface of the speleothem, which forms the fluid inclusion water when it is eventually trapped, may be affected by evaporation, which would lead to increasing $\delta^{18}O$ and $\delta^2H$ values of the water. However, due to the very long time-constant of oxygen isotope exchange between water and calcite, it takes hours to imprint the oxygen isotope signature of the (evaporated) water on the dissolved bicarbonate. Thus, in case of evaporation inside the cave, the $\delta^{18}O$ values of the precipitated calcite would still




roughly reflect the $\delta^{18}O$ value of the original drip water, whereas the trapped fluid inclusion water may be strongly affected by evaporation and exhibit elevated $\delta^{18}O$ values (Fig. 3). These processes have been modelled by Deininger et al. (2012), who
predicted an enrichment of the $\delta^{18}O$ values of the water film on the surface of a stalagmite of several per mil due to evaporation, whereas the effect on the $\delta^{18}O$ values of the precipitated calcite was below 1 ‰.

Ventilation in Larga Cave is mainly driven by temperature differences between the atmosphere and the cave (Vieten et al., 2016). During the dry winter season, over the course of the day, the part of the cave hosting the studied speleothem is warmer than the atmosphere inducing a buoyancy-driven air flow. According to our results, evaporation effects are strongest during
the comparably cooler and drier periods of the record. This in turn suggests that cave ventilation was particularly pronounced in these times, which would argue for a stronger seasonal temperature contrast during stadial intervals. This observation seems to be consistent with the hypothesis of enhanced northern-hemisphere seasonality during abrupt cooling events (Denton et al., 2005). Similarly, Ziegler et al. (2008) reported persistently high summer temperatures of the Atlantic warm pool also during north Atlantic cold events, such as Heinrich stadials, suggesting that these seem to influence mainly the winter climate
conditions in the Caribbean. It is therefore plausible that a pronounced and persistently strong seasonal temperature difference with particularly cool temperatures during the winter season enhanced winter type ventilation conditions in Larga cave. Hence, ventilation, and thus, the occurrence of evaporation on the stalagmite's surface would have been significant.

### 4.1.3    Applicability of the oxygen isotope equilibrium thermometer

The previously discussed processes have implications for the applicability of the oxygen isotope equilibrium thermometer
using fluid inclusion and carbonate $\delta^{18}O$ values from PR-LA-1. Post-depositional enrichment of $^{18}O$ in fluid inclusion water due to evaporation apparently results in high $^{18}\varepsilon_{c/f}$ fractionation values between fluid inclusion water and calcite, because such influenced samples do not preserve the original $\delta^{18}O_f$ signal. Samples plotting off the MWL are thus discarded from further inferences. Note, evaporation effects would increase reconstructed temperatures by a factor of two to even three, hence, temperature reconstruction is very sensitive to evaporation, which must be considered when interpreting meaningful
temperature values as likely being maximum temperature estimates. Figure 4 shows that also samples, which plot on the GMWL in Fig. 3, might show signs of evaporation. This is, e.g., the case for the sample taken from Heinrich Stadial (HS) 1 (15.9 ka). Here, other paleoclimate records as well as the PR-LA-1 calcite $\delta^{18}O$ values and elemental proxies suggest that this phase was the coolest and driest in Puerto Rico and the Caribbean realm (Warken et al., 2020;Escobar et al., 2012;Grauel et al., 2016). The temperature calculated from the PR-LA-1 fluid inclusion value, however, would be considerably warmer than
at present-day. As outlined above, we have to assume that evaporative effects also influenced speleothem and fluid inclusion formation during HS 1, shifting the $\delta^{18}O_f$ to heavier (more positive) values. This demonstrates that samples may be influenced from evaporation despite a position on or close to the MWL. We therefore consider all calculated temperatures as a maximum estimate, and results lower than +3°C compared to modern temperatures as in first order plausible.



### 4.2 Reconstruction of hydro-climate and paleo-temperatures in the western tropical Atlantic region

#### 4.2.1 Fluid inclusion δ¹⁸O values suggest reduced convective activity during the last glacial

Despite their position close to the MWL, $\delta^{18}O_f$ values in cluster (1) are significantly higher by +2.6 ± 1.3 ‰ compared to modern drip water $\delta^{18}O_w$. At the time of deposition of speleothem PR-LA-1 (46.2 to 15.3 ka), the $\delta^{18}O$ signature of the (sub)tropical Atlantic surface water was higher by about 1.5‰ due to larger global continental ice volume and increased salinity of the western tropical Atlantic (Hagen and Keigwin, 2002;Lea et al., 2003;Schmidt and Spero, 2011). Consequently,

glacial drip water represented by fluid inclusion $\delta^{18}O_f$ values must be corrected for this 1.5‰ offset from present-day. Millo et al. (2017) reported a similar enrichment of about 4‰ compared to modern drip water in last glacial fluid inclusion water in a subtropical stalagmite from Brazil, and a similar glacial-interglacial offset has although been observed in $\delta^{18}O$ values of stalagmites from Guatemala (Winter et al., 2020) and Florida (van Beynen et al., 2017).

We interpret the offset in cluster (1) to reflect a generally weaker convective activity during the last glacial period compared

to modern conditions and associated reduced summer rainfall (e.g., Arienzo et al., 2019;Winter et al., 2020;Lases-Hernandez et al., 2019;2020;Bird et al., 2020). Today, Puerto Rico rainfall is isotopically enriched when it is formed by occasional northerly cold fronts as well as orographic processes of trade winds (Scholl and Murphy, 2014). In contrast, isotopically lighter precipitation is associated with convective rain, such as hurricanes and tropical storm events. The type of precipitation, i.e., convective vs. orographic, depends to some extent on the meridional location of the ITCZ as well as related Caribbean SSTs

(Winter et al., 2020). In this study, the mean of the reconstructed fluid inclusion temperatures for the LGM (c. 25 – 18ka) is 19.4 ± 3.3 °C. Since our reconstructed temperatures can be regarded as upper limits, due to an underlying imperceptible amount of evaporation, this supports generally cooler temperatures than today over the course of the LGM. The observation of a glacial-interglacial temperature difference of at least 3°C highlights the results of previous Caribbean SST records (Rama-Corredor et al., 2015;Lea et al., 2003;Schmidt et al., 2006b). Such low temperatures were presumably too cold to sustain strong

convective activity during the LGM. Accordingly, Winter et al. (2020) suggested that deep atmospheric convection during the Holocene was triggered by exceeding a SST threshold of 27.5°C in the tropical ocean leading to a shift of the hydro-climatic regime from a drier state and weak convection during the last glacial to a humid state and strong convection after 9 ka over Central America.

The mean value of Puerto Rican fluid inclusion temperatures obtained during MIS 3 (45.5 - 29 ka) is 20.4 ± 2.6 °C, which is

on average 1°C warmer than the LGM. Despite large uncertainties, this is consistent with a late MIS 3 warmer than the LGM in the Guiana Basin (Rama-Corredor et al., 2015) and Brazil (Millo et al., 2017). For Central Mexico, Caballero et al. (2019) report a mean annual temperature of MIS 3 comparable to the present-day. From a global perspective, MIS 3 was c. 2°C warmer than the LGM, due to a northern hemispheric summer insolation maximum and resulting smaller ice sheets (Van Meerbeeck et al., 2009;Svendsen et al., 2004). According to Rama-Corredor et al. (2015), western tropical Atlantic SSTs were

close to the threshold of 27.5°C during certain intervals of MIS 3 (Fig. 5), allowing sustainment of convective activity at least temporarily and/or over a longer time interval over the course of the year. Consequently, relatively wet environmental




conditions - even though still drier than those of the Holocene – are supported by records from Mexico, Guadeloupe and the Bahamas (Lozano-García et al., 2015;Royer et al., 2017;Arienzo et al., 2017).

**Figure 5: Paleo-temperatures and hydro-climate in the western tropical Atlantic region during the LGM and MIS2. From top to bottom: (a) Cariaco basin reflectance record representing the mean meridional position of the ITCZ over northern south America (Deplazes et al., 2013); (b) Speleothem PR-LA-1 $\delta^{18}O_c$ values from Warken et al. (2020), representing local high-resolution hydroclimate; (c) Guiana basin Uk37 sea surface temperature reconstruction (Rama-Corredor et al., 2015); (d) Ostracod $\delta^{18}O$ derived temperature record from Lake Petén Itzá (Guatemala) (Escobar et al., 2012); (e) Speleothem fluid inclusion based**
**temperature reconstruction from the Bahamas (Arienzo et al., 2015); (f) Fluid inclusion derived temperatures from PR-LA-1 (this study). The horizontal dashed line indicates modern cave air temperature at Larga cave. (g) green triangles indicate the timing of Larga cave ventilation events as suggested by fluid inclusion stable isotopic composition influenced by evaporation. In the tropical Atlantic, Heinrich stadials (HS1 to 4) as well as Greenland stadials are associated with cold and dry conditions (light blue bars), and Greenland interstadials are associated with warmer and more humid climatic conditions (light red lines).**



### 4.2.2    Millennial-scale climate variability in the western tropical Atlantic

On the millennial scale, fluid inclusion $\delta^{18}O_f$ (and $\delta^2H_f$) values exert pronounced swings between very high values of up to 2‰ (30‰) during stadial phases, and down to 0‰ (15‰) during interstadial phases between 45 and 27 ka. The pattern largely follows the evolution of the high-resolution calcite stable isotope record (Fig. 2). Vertical blue bars in Fig. 2 and Fig. 5 illustrate the intervals of elevated $\delta^{18}O_c$ values during Heinrich stadials, which are associated with relatively cold and dry conditions in the circum-Caribbean area. The calculated upper limits for paleo-temperatures during HS2 and HS3 are between 16.5 ± 1.4 °C and 21.4 ± 1.4 °C and between 17.6 ± 1.7 °C and 18.7±1.4 °C, respectively (Fig. 5, Table S2). Accordingly, we estimate a mean, minimum cooling compared to the present cave air temperature of -2.9 ± 2.6 °C for HS 2 and -4.4 ± 2.2 °C for HS 3. These values are consistent with the results of Arienzo et al. (2015), who found an average temperature decrease of 4°C across Heinrich stadials 1 to 3 for the Bahamas. Many studies document that HS 1 exhibited the most pronounced cooling of all Heinrich stadials (Arienzo et al., 2015;Grauel et al., 2016). HS1 was also more pronounced in the isotopic and elemental calcite signals of our sample compared to HS2 or HS3 (Warken et al., 2020). Escobar et al. (2012), and Grauel et al. (2016) found an even stronger cooling down to 10°C for HS1 in Lake Petén Itzá, Guatemala (Fig. 5). We note, however, that these studies used bottom dwelling ostracods, which may have recorded winter – biased lake temperatures. In addition, the strong maritime climate of Puerto Rico located in the eastern Caribbean makes less pronounced temperature variations appear reasonable. In addition to absolute temperature estimates, the samples which show signs of evaporation from cluster (2) suggest that during these times an increased seasonal temperature difference between persistently warm summer temperatures but cooler winter temperatures lead to enhanced cave ventilation. The timing of these samples which includes not only the Heinrich stadials but also Greenland stadials and certain intervals of the LGM is highlighted in Fig. 5. This further illustrates that our dataset suggests that enhanced seasonality may have occurred during these intervals, which would be indicative of a persistent summer expansion of the Atlantic warm pool, but relatively cool winter temperatures.

Besides the temperature drops associated with Heinrich stadials, the expression of Greenland interstadials is still not well constrained in the American tropics (Warken et al., 2019). The mean temperature calculated from samples associated with interstadial events in speleothem PR-LA-1 is 22.0 ± 1.8°C (samples 15, 24, 28, 35), which is in the range of present-day cave temperature, and on average 1.6°C warmer than the MIS 3 mean temperature. In comparison to average stadial temperatures of 18.6 ± 1.1°C (samples 16, 18, 19), this indicates a difference in the range of 3.4 ± 2.8°C between stadial and interstadial phases in Puerto Rico. In the tropical Guiana Basin at 7°N latitude, abrupt glacial SST variations on the millennial scale are in the range of about ±1°C (Rama-Corredor et al., 2015). Further north, Them Il et al. (2015) report a shift of c. 2°C associated with stadial/interstadial transitions in the Florida Strait (24°N), whereas shifts by 2 to 5°C have been observed in subtropical SSTs, such as at the onsets of GI8 or 12 at the Bermuda Rise at 33°N (Sachs and Lehman, 1999). The available temperature reconstructions thus form a north-south transect and suggest a progressive attenuation of millennial-scale temperature oscillations towards the equator.



Our results thus show that the interstadial phases in the western tropical Atlantic were associated with relatively warm temperatures, which may have been even comparable to present-day. Such warm temperatures favoured convective activity,
and thus, higher precipitation amounts in the region. Marine reconstructions showed that interstadials were associated with reduced upper water column salinity, while stadial events were associated with increased salinity (Them Il et al., 2015). These observations support our results that meridional shifts in the position of the ITCZ in response to changes in the strength of the AMOC affected the evaporation-precipitation ratio in the western tropical Atlantic (Schmidt and Spero, 2011;Arienzo et al., 2017;Peterson et al., 2000;Schmidt et al., 2006a;Warken et al., 2020).

## 395 5 Conclusions

We demonstrate that the stable isotopic composition of fluid inclusions can be strongly influenced by evaporative effects. In Larga Cave, the temperature driven ventilation potentially favoured such conditions, and our data suggest that this effect may have been especially pronounced during cold and dry conditions, such as during Heinrich and Greenland stadial phases and parts of the LGM. During warmer Greenland interstadials, most samples suggest that isotopic modification due to evaporation
is negligible. In Larga cave, this process is apparently climate-related, making it possible to omit the potentially affected sub-samples for paleo-temperature calculation.

Our dataset thus provides valuable information about last glacial hydro-climate and temperature variability in the western tropical Atlantic and allows to constrain climatic changes on the millennial to orbital scale. For the presumably coldest phases of the last glacial, the Heinrich stadials, we observe a minimum cooling of 2.9 ± 2.6 °C during HS 2 and of 4.4 ± 0.6°C during
HS 3. In contrast, during interstadial phases, temperatures were on average 22.0 ± 1.8°C, which is comparable to modern values. The analysis of stable water isotope values in speleothem fluid inclusions further suggests that, compared to present-day, the precipitation regime in the western tropical Atlantic was characterized by generally weaker convective activity during the glacial period as compared to present-day. Especially during the LGM, average temperatures were at least 3°C cooler, and thus too cold to sustain deep atmospheric convection. The intermediate background climate state during MIS 3, with average
temperatures about 1°C warmer than the LGM, favoured an increased sensitivity of western tropical Atlantic hydro-climate to perturbations of the AMOC and northern hemispheric temperature changes on the millennial scales. In this period, the region experienced significant shifts between a slightly weaker – but still active - convective regime during the warmer interstadial phases and a dry/non-convective regime during the colder stadial phases.

## Data availability

The results of fluid inclusion isotope analyses from this study are available in the appendix to this publication. The in-house Python code for the evaluation of the raw data is available under https://github.com/bhemmer/IsoFluid



**Author contributions**

Conceptualization: SW, TW, DS, HV; Formal analysis: TW, SW; Funding acquisition: TK, NF, MS, AW, DS; Investigation:
TW, SW; Methodology, Resources: TW, TK, MS, NF; Visualization: TW, SW; Writing – original draft preparation: TW, SW
Writing – review & editing: SW, TW, TK, HV, DS, RV, MS, AW, NF

**Competing interests**

The authors declare that they have no conflict of interest.

**Acknowledgements**

N. Büttner, M. C. Juhl and R. Eichstädter are thanked for help in the lab and with sample preparation. B. Hemmer is thanked
for support in constructing the Python evaluation code. We further thank W. Aeschbach for ongoing support. T.W. and T.K.
acknowledge funding from DFG grant KL 2391/2–1, and support by the Heidelberg Graduate School of Fundamental Physics
(DFG-GSC129). This project further benefitted from infrastructure supported by grant DFG-INST 35_1143-1 FUGG and
DFG-INST 35/1270-1 FUGG (to N.F.), as well as DFG research grants DFG SCHO 1274/6-1, SCHO 1274/9-1, SCHO
1274/11-1 (to D.S.), and grant number ATM-1003502 (to A.W.) from the National Science Foundation (NSF).

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
