# Peer review of "Last glacial millennial-scale hydro-climate and temperature changes in Puerto Rico constrained by speleothem fluid inclusion $\delta^{18}O$ and $\delta^{2}H$ values"

_Climate of the Past, 2021_

## Author Comment (AC1)

Dear Climate of the Past Editorial Board,

Dear reviewers,

All the authors would like to thank the two anonymous reviewers for their time spent on reading and evaluating the manuscript, and for their constructive comments on our study. The suggested changes and additions will significantly improve the quality of our manuscript and are highly appreciated by all contributors. Below, we have listed our responses to all comments and the changes made in the manuscript. Here, the comments of reviewers are printed in black, the author responses in blue.

All contributors of this study have stated their agreement with the following changes and additions.

Kind regards,

Sophie Warken, on behalf of all authors

**Response to Anonymous Referee #1:**

However, I see two main problems with the data.

   1) The evaporation process.
The inclusion data clearly reflect evaporation. Evaporation may occur in the atmosphere and on the surface. In this case the precipitating calcite can reflect the shifted water δ18O value, provided that increasing evaporation is not associated with increasing temperature, whose fractionation effect can counteract the evaporation related isotope shift, leading to constant δ18O in the calcite. The present study suggests cold and dry conditions, in which case the positive δ18O change in the water would be enhanced in the calcite due to the elevated calcite-water oxygen isotope fractionation, which is obviously not the case
Another possibility is evaporation along the karstic water migration route. In this case evaporation would also be associated with enhanced CO2 degassing and strong PCP. Warken et al. (2020) excludes this possibility as the Sr/Ca ratios are not changing along with the Mg/Ca values. Additionally, the water arriving at the stalagmite surface would already be evaporated and its isotope compositions would be shifted, that should appear in the calcite.
The next possibility is evaporation on the stalagmite surface and this is what the authors suggest as a potential process. However, it is difficult to imagine that calcite is precipitated from an unaffected solution, then the water film is evaporated, and the calcite precipitating from the evaporated water would no show any δ18O shift.
We thank the reviewer for the detailed comment on the evaporation process and for the opportunity to clarify our explanations regarding this aspect. Even though Reviewer 2 acknowledged our model as well explained, the description of the proposed process was apparently still not clear enough in the previous version of the manuscript.
Firstly, we want to emphasize that we do not discuss evaporation above the cave, but only on the stalagmite surface, which we state at several locations in the manuscript. Thus, we would refrain from including a detailed discussion of the effects of evaporation in the atmosphere and the karst aquifer above the cave in the revised MS. Instead, the key point is the slow oxygen isotope exchange during the evaporation process on the stalagmite surface. While calcite precipitation occurs within seconds to tens of seconds, the oxygen isotope exchange between water and $HCO_3^-$ evaporation takes much longer (e.g., Scholz and Dreybrodt, 2011, and references in previous version of manuscript). Therefore, if the water is affected by evaporation, the $δ^{18}O$ signal will not be imprinted in the precipitated calcite, because this will have already precipitated. To explain this, we have stated in L267ff the previous version of the manuscript: *"When the drip water [...] falls onto the surface of a stalagmite, degassing of excess $CO_2$ occurs within a few seconds (Dreybrodt and Scholz, 2011) with a negligible effect on the $δ^{18}O$ value of the dissolved bicarbonate. Then, precipitation of $CaCO_3$ starts immediately, and the $δ^{18}O$ value of the bicarbonate that was previously equilibrated with the drip water is imprinted on the $δ^{18}O$ value of the precipitated calcite."*

The above-described process implies that the majority of calcite is precipitated soon after a certain super-saturation has been sur-passed and nucleation triggered. The Ca concentration of the solution then drops exponentially and only a minor component can still be precipitated from the remaining solution (see e.g., (Dreybrodt and Scholz, 2011;Deininger et al., 2012;Scholz et al., 2009;Mühlinghaus et al., 2007)). Given the smaller amount of Ca typically left in the solution even under evaporative conditions, only a minor amount of calcite is formed that will not significantly influence the bulk isotope values. However, the water that is archived after closure of water in the cavities is an aliquot of the evaporated solution. To illustrate this process, we provide a cartoon in Fig. R1.

[Figure]

*Figure R1: Graphical illustration of the post-depositional/pre-entrapment evaporation process.*

To improve the explanations in the revised manuscript, we will adapt section 4.1.2 describing the "Climate-related evaporation effects on fluid inclusion isotope values in Larga cave". We suggest to change the paragraph from L267ff in the previous version as follows (changes with respect to previous manuscript marked in green):

*"When the [...]. We note that oxygen isotope fractionation during precipitation of speleothem calcite is a complex process (e.g., Dreybrodt and Scholz, 2011;Guo and Zhou, 2019;Hansen et al., 2019;Scholz et al., 2009), and oxygen isotope fractionation does in most cases not occur under conditions of oxygen isotope equilibrium. After rapid precipitation of speleothem calcite, the drip water remaining on the surface of the speleothem, which forms the fluid inclusion water when it is eventually trapped, is progressively affected by evaporation, which leads to increasing $\delta^{18}O$ and $\delta^2H$ values of the water, and finally to fluid inclusions that are not in isotope equilibrium with the host-calcite. However, due to the very long time-constant of oxygen isotope exchange between water and bicarbonate, it takes hours to imprint the oxygen isotope signature of the (evaporated) water on the dissolved $HCO_3^-$, and hence, the precipitated calcite. In addition, since the Ca concentration of the drip water drops exponentially, only a minor component of calcite could still be precipitated from the remaining solution (Scholz et al., 2009;Mühlinghaus et al., 2007;Deininger et al., 2012;Dreybrodt and Scholz, 2011). Given the smaller amount of Ca typically left in the solution, even under evaporative conditions only an insignificant amount of calcite may be formed from the solution affected by evaporation. Thus, in case of evaporation inside the cave, [...]."*

Please also note our responses to the following, related comments, and our suggestions for a revised manuscript.

Further, the low H2O contents should be associated with higher δ13C values in the calcite. I tried to plot the water contents on the δ13C record presented by Warken et al (2020), and the relationship is not convincing. I plotted the data and it is apparent that the water content and the fluid inclusion δ2H and δ18O data are negatively correlated (with r2 values around 0.32 for logarithmic fitting). However, it is

not clear why the evaporation from the water film on the surface would result in lower amount of fluid inclusions in the calcite. Evaporation should increase the calcite saturation, leading to faster carbonate precipitation and more inclusions trapped.

We thank the reviewer for this comment. In fact, the water content has only a weak anti-correlation with the averaged calcite $\delta^{13}C$ values (r = -0.37, p <0.05). Laboratory experiments have shown that the enrichment of calcite $\delta^{13}C$ values is much larger than the enrichment of calcite $\delta^{18}O$ values with increasing residence times, which is however due to prior calcite precipitation (PCP) (Hansen et al., 2017). Thus, we are not aware of other processes which would lead to a strong anti-correlation of the water content and calcite $\delta^{13}C$ values.

We further agree with the reviewer, that there is a correlation between the water content and the fluid inclusion stable isotope values. We suggest to add an additional statement to this aspect to the revised version of the manuscript to section 3.1 (corresponding to L168ff previous manuscript): "*The relationship between fluid inclusion stable isotope values and water content can be also expressed numerically with a negative correlation of $\delta^{18}O_f$ and the measured water content with r = -0.72 (r² = 0.51) for logarithmic fitting (Figure S 2).*". We note here also, that for a linear fit, r = -0.55 for $\delta^{18}O$, and -0.64 for $\delta D$.

We hypothesize that the relation between water content (amount and/or volume of fluid inclusions) and evaporation mirrors the reduction in total water volume of the inclusion due to evaporation. This process can be described by a Rayleigh fractionation model, as shown in Fig. R2, which we will also be added to the supplementary material of the revised manuscript. We will also include the following statement to section 4.1.2 (corresponding to L254ff in the previous version): "*Calculation of the evolution of both water content and fluid inclusion $\delta^{18}O$ values with a Rayleigh fractionation approach (**Fehler! Verweisquelle konnte nicht gefunden werden.**) supports this interpretation and shows that a moderate evaporation of 50-75% of the initial water content (between 0.5 and 3µl/g) is sufficient to explain the observed enrichment of more than 12‰ of the fluid inclusion $\delta^{18}O_f$ values.*"

[Figure]

*Figure R2: Fluid inclusion stable isotope data (represented by δ18O values) vs. measured water content. The data suggest an exponential enrichment of $\delta^{18}O$ values with decreasing water content (r = -0.72 for logarithmic fitting). Solid lines indicate Rayleigh fractionation with different initial water volumes $V_{ini}$ and an equilibrium oxygen isotope fractionation of 9.3‰ (Majoube, 1971). Since the kinetic fractionation factor under evaporative conditions is much higher, we also show the Rayleigh fractionation curves for a fractionation factor of 18‰. These curves demonstrate that the evolution may be explained by progressive evaporation of the water in the fluid inclusions before closure resulting in an associated enrichment of the heavy stable water isotopes.*

An alternative process is partial leaking of inclusions. As the sample chips were held in a carrier gas flow, they were most probably heated to remove the adsorptively bound water. Although this is not mentioned exactly, the authors mention 120 °C in the vacuum line to avoid water condensation. Heating speleothems to 120 °C may partially open large inclusions. The changes in climate conditions may change the carbonate fabric, inclusion entrapment, and hence the tendency of inclusion leakage during heating, as well. Partial removal of inclusion-hosted water due to inclusion leaking may lead to lower water content and shifted isotope compositions. This process is also supported by the large variations in water contents and isotopic compositions in the same sample (e.g. sample 24). Heating the samples at different temperatures (80, 100, 120 °C) in a vacuum system and record the vacuum achieved, or measuring pieces from the strongly evaporated laminae at lower temperature (e.g., 80 °C) would provide means to investigate this potential measurement bias.

The reviewer is absolutely right that such an effect may occur. However, in order to see if this effect is indeed a problem, we have always monitored the water mixing ratio in the gas stream through the isotope analyser after insertion of the sample pieces. We never observed anomalies, such as sudden increases in the water vapour concentration (e.g., by cracking of large inclusion), nor corresponding changes in the isotopic values of the background water vapour. Samples, which yielded evaporated water isotope values, behaved similarly as samples that yielded non-evaporated values. For additional illustration, Fig. R3 shows the time every sample was heated prior to crushing. If the evaporation and/or the measured water content were related with the heating in the fluid inclusion line, we would expect a correlation between the heating time and the isotope values and/or the water content. As the attached plot shows, this correlation is not given.

We further have the experience with samples from different studies that when anomalous water backgrounds during warm-up are evident, this occurs particularly in diagenetically altered samples, and alteration is not apparent in our sample material, as is evident from thin section photographs (Figure R4, will be added to revised supplementary material). To clarify this detail also for other readers of the paper, we will add corresponding statements in the revised version of the manuscript both in the methods section and in the discussion:

- We suggest to add *"Microscopic inspection of thin sections shows that PR-LA-1 consists of calcite in columnar fabric and shows no signs of dissolution and/or recrystallization (Figure )."* to section 2.1. (corresponding to L95 in the previous MS)

- We suggest to add *"After connecting the sample, the setup requires approximately 90 minutes until the water vapour background has reached stable conditions. When the standard deviation of the water vapour concentration is below 20 ppmV over 30 minutes, the sample is crushed with a hydraulic crusher."* to section 2.2. (corresponding to L127 previous MS)

- Finally, we will add *"Isotope effects due to partial evaporation of fluid inclusion water can in principle also occur during the warm-up phase in the crusher when fluid inclusions are insufficiently sealed to withstand the pressure that builds up during warm-up. Based on our experience, water loss in the warm-up phase causes noticeable disruptions in the water background of the analyses, but this has been observed mostly for diagenetically compromised samples so far. We note that the water vapour background was continuously monitored before crushing, and no anomalies were observed for the samples from PR-LA-1 (Weißbach, 2020). Figure also shows no clear relationship between shape and distribution of fluid inclusions in PR-LA-1 nor do we see signs of diagenetic alteration in stalagmite PR-LA-1, which could have favoured inclusion leaking during heating.*" to section 4.1.1 (corresponding to L235ff previous MS)

[Figure]

*Figure R3: Fluid inclusion analysis data (water content (left) and δ¹⁸O (right)) with respect to the time between insertion of the sample chip in the heated fluid inclusion line and crushing. No systematic relationship between the time before crushing and the evaporation pattern in the fluid inclusions can be deduced.*

[Figure]

*Figure R4: Transmitted light photographs of polished thin sections of stalagmite PR-LA-1 (obtained with a Keyence VHX-6000 digital microscope) showing typical shapes and distribution of fluid inclusion. The photographs were taken from sections corresponding to depths of analysed fluid inclusion samples, where the red arrow shows the respective growth direction. (a) and (b) show examples of thin, elongated fluid inclusions from sections corresponding to samples 1, as well as 5 and 5b. (c) and (d) are examples of fluid inclusion rich parts corresponding to the depths of samples 19 and 20. (e), (f) and (g) are examples for sections with lower water content, and correspond to samples depths 21b, 22 and 23, respectively. (h) and (i) show examples of less elongated fluid inclusions corresponding to sample depths 27 and 27b, respectively. Lack of evidence of dissolution and/or recrystallization of calcite crystals suggests that fluid inclusions are pristine. The isotopic composition of fluid inclusion samples 1 (a), 21b (e), 27 (h) and 27b (i) show signs of evaporation, in contrast to samples 5b (b), 19 (c), 20 (d). In samples 22 (f) and 23 (g) the water content was too low for fluid inclusion analysis. j) Image of a whole thin section of PR-LA-1, illustrating the dominant fabric with large columnar calcite crystals. Petrography shows that no clear relationship can be found between shape and distribution of the fluid inclusions and the isotopic composition of the inclusion water.*

Additionally, calcite and fluid inclusion petrography is essential in order to see if there is any fabric and textural change related to changes in climate conditions.

We agree with the reviewer and will add thin section photographs of several sections, where samples for fluid inclusion analysis were taken (Figure R4, will be added to the revised supplementary material). We will also add a corresponding statement in the revised manuscript at L95ff and L245ff (compare previous comment).

Further, the fluid inclusion data should be compared with the averaged δ13C, δ18O and Mg/Ca values in order to see if these evaporation- and degassing-sensitive proxy data are affected. Enhanced cave ventilation is mentioned in line 372 that should be supported by δ13C-δ18O correlation.

Since we explain the deviation of fluid inclusion $\delta^{18}O$ and $\delta D$ values from the meteoric water line by a post-depositional-pre-entrapment effect, it should not generally be related to the calcite proxy signals, as explained in the previous comments. Fig. R5 shows the comparison between the averaged calcite proxies and the fluid inclusion data (water content, and $\delta^{18}O$ values). No significant relationship between the calcite and fluid inclusion proxies can be obtained, which supports our previous explanations. In addition, we will include a plot of the speleothem calcite proxies ($\delta^{18}O$, $\delta^{13}C$, and Mg/Ca) in the revised supplementary material (Figure R6).

[Figure]

*Figure R5: Fluid inclusion data compared to averaged calcite $\delta^{13}C$, $\delta^{18}O$ and Mg/Ca data (Warken et al., 2020).*

[Figure]

*Figure R6: Comparison of PR-LA-1 proxies including (from top to bottom) the growth rate (black) as well as calcite δ¹⁸O (dark red), δ¹³C (dark grey), and Mg/Ca (magenta) values (Warken et al., 2020). Lines indicate the unsmoothed data, symbols the averaged values for the depth range covered by the fluid inclusion samples, respectively. Fluid inclusion data include δ¹⁸O_f (red), δ²H_f (orange), and the measured water content (blue). Symbol colors indicate the samples with high (dark colored symbols) and low (light colored stars) water content with a threshold value of 0.55 μl/g, respectively. The bottom-most panel shows the temperatures from the fractionation between fluid inclusion and speleothem calcite δ¹⁸O values. Yellow colors indicate temperatures which are regarded as not reliable, while green symbols indicate reasonable paleo-temperatures, as discussed in the main manuscript.*

2) Paleotemperature uncertainty. The paleotemperature data are valuable, but their scatter is too high to discuss <2 °C differences between different periods. The authors state that the interstadial values are 1.6 °C higher than the MIS3 average. However, the interstadial average contains data from sample 24, which is very inhomogeneous. As the authors mention, some of the data were excluded from the calculations due to the evaporation affect, and the temperatures are taken as maximum values. However, if some periods are affected, and some others are not (due to carbonate texture differences), then the uncertainty becomes much higher that the 1.6 °C difference.

We acknowledge the uncertainty, and will revise the manuscript accordingly. In particular, we will remove the statement that "interstadial values are 1.6°C higher than the MIS3 average". We also suggest to change L334ff (previous version of MS) to: *"The mean value of Puerto Rican fluid inclusion temperatures obtained during MIS 3 (45.5 - 29 ka) is 21.7 ± 2.6 °C. Despite the relatively large uncertainties, this may support a warmer MIS 3 than LGM, which has been found in the Guiana Basin (Rama-Corredor et al., 2015) and Brazil (Millo et al., 2017)."*

The authors state that the 22.0 ±1.8 °C temperature for the interstadial periods is comparable with the present day condition. However, the former is obtained from inclusion data, the latter is from direct temperature measurement. I know that it is not a nice suggestion to collect more samples from Puerto Rico, but if the authors accidentally have modern samples or at least Holocene stalagmites, only some measurements may strengthen this statement.

We agree with the reviewer that it would be nice to support the dataset with additional data from recent or Holocene sample material from the same cave. Unfortunately, there is no sample material from the same cave chamber available to us at the moment. Test measurements from recent material from other parts of the cave were not successful due to low water contents.

**Addional comments:**

line 155: „Sections with a relatively high water content, e.g., around c. 28 or 34ka...". The water content at 34 ka is low, as far as I see.
Statement will be changed to […] around 24 and 28ka […].

line 194: It would be informative to plot curves of the Johnston et al. (2013) equation for different temperatures in Fig. 4a.
We thank for this valuable suggestion. Since we will change to the water–calcite oxygen isotope fractionation relationship after Tremaine et al., 2011 (compare comment of Reviewer 2), we will include the so-derived curves for cave temperatures of 15, 20 and 25°C, respectively.

line 295: „Post-depositional enrichment of 18O in fluid inclusion water due to evaporation". Post-depositional but pre-entrapment?
Yes. We will specify this (and similar statements) in the revised manuscript.

line 302: In order to support this sentence, the fluid inclusion data and the proxy records should not only plotted together, but a statistical analysis between these variables (with the high-resolution proxy data averaged for the inclusion sampling) is suggested
We are confident that the cited literature does sufficiently support the statement that "*HS1 was the coolest and driest period in the Caribbean, including Puerto Rico*".

**Response to Anonymous Referee #2**

This manuscript presents a study of fluid inclusion isotope composition from a speleothems sampled in a cave from Puerto Rico, which today experiences seasonal ventilation variations. The data in this paper are high quality fluid inclusion dD and d18O measurements covering the 46-15 ka period made using state of the art methodology.
The excellent team of authors show that the FI data can grouped into two distinct d18OdD groups. One group falls on the global and local meteoric water lines. This group represents samples FI yield greater than 0.55 microliter/g. The second group, characterized by FI yields less than 0.55 microliter/g shows a strong deviation from the local meteoric water line in an apparent evaporation trend characterized by a dD/d18O slope significantly lower than that of the meteoric water line. The calcite of former group is proposed to form at, or near, temperature d18O equilibrium with the fluid inclusion water.
The calcite d18O of the second group is similar in range to the first group, and consequently is in isotopic disequilibrium with the FI waters of the second group. The controls of the proposed evaporation process are climatic-driven ventilation during colder periods. Thus, the first group of FI waters and calcites are considered to reflect high cave humidity warm period conditions, whereas the evaporated samples reflect colder periods characterized be enhanced cave ventilation.
The kinetic mechanism proposed to account for these contrasting modes is very interesting. The speleothem calcite precipitates rapidly in both warm and cold periods in or near temperature isotopic

equilibrium with cave drip waters. In the colder periods characterized by enhanced ventilation, evaporation of the dripwater films and the fluid inclusions trapping these evaporated waters occurs, yielding the deviations observed from the meteoric water line. However, the kinetics of the calcite-bicarbonate exchange is so slow that the speleothem calcite does not reset to the evaporated water values: the calcite thus retains the original 'equilibrium precipitation signature' attained prior to the onset of evaporation driven d18O changes. The model is well explained and the findings of the paper are both interesting and innovative.

We thank the reviewer for her/his supportive comments. We are also glad to hear that our model is well explained and considered as interesting and innovative, in particular because reviewer 1 had several questions about the suggested process.

This is an intriguing experimental data set, well-written, innovative in scope, and the manuscript certainly deserves publication. Nevertheless, there are several points that I would like to see addressed before the manuscript becomes published.

What is the advantage of using the Johnson et al (2013) thermometer calibration over the more generally used Tremaine et al (2011) calibration. The Tremaine et al calibration has become something of a standard in cave thermometry.
We agree with the reviewer. In the revised version of the manuscript, we will use the Tremaine et al. (2011) temperatures for interpretation, which shows only slight differences to the Johnston et al. (2013) equation.

I have some reservation over the explanation of the proposed evaporation trend. The shift of d18O from the meteoric water line value is as much as 12 permil. This is a huge evaporation effect! The authors should use a Rayleigh calculation to estimate fraction of the original water that has evaporated away before leaving the residual water in the fluid inclusions of the evaporated samples. I would guess that the amount of residual water in the fluid inclusion is a small fraction of the calculated original water amount, and that the calculated original water amount is much higher than the water present in the first group samples, which contain about 0.5-2 microliter/g.
Fig. R7 shows a Rayleigh calculation, which demonstrates that substantial evaporation is needed to reach a change of 12 ‰. For example, 75% are required if the equilibrium fractionation factor between water and water vapour at 25°C is used. For an initial water volume of 2µl/g, still 0.5 µl/g would be left. If the initial water volume was 0.5 µl, about 0.125µl would be left at a Rayleigh fractionation of 9.3 ‰ (Majoube, 1971). However, the kinetic fractionation factor under evaporative conditions is much higher. For instance, at a fractionation factor of 18‰, an evaporation of only 50 % is sufficient to reach 12‰ enrichment. Fig. R2 (in response to the comments of reviewer 1) shows that the measured fluid inclusion data (water content and stable isotope values) can be explained by this Rayleigh fractionation approach. In summary: the maximum evaporation before inclusion sealing is not extreme and somewhere between 50 and 75%; the initial water volume in the evaporated samples does not exceed the first group but generally stays below, even after accounting for the evaporation.
To clarify this also in the revised manuscript, we will add this aspect to the discussion in section 4.1.2 (corresponding to L254ff in the previous MS): "*Calculation of the evolution of both water content and fluid inclusion $\delta^{18}O$ values with a Rayleigh fractionation approach (Figure S 2) supports this interpretation and shows that a moderate evaporation of 50-75% of the initial water content (between 0.5 and 3µl/g) is sufficient to explain the observed enrichment of more than 12‰ of the fluid inclusion $\delta18Of$ values.",* and will include Figure R2 as Figure S2 in the revised supplementary material.

[Figure]

*Figure R7: Rayleigh-fractionation curve showing the evolution of the fluid inclusion $\delta^{18}O$ value in the evaporating water relative to the remaining fraction (f).*

The authors should explain how their proposed mechanism of fluid inclusion entrapment is compatible with high degrees of evaporation. Is there any alternative mechanism that could account for the apparent evaporation trend e.g, a process during analytical handling?

As outlined in the response to the previous comment, evaporation of 50-75% is sufficient to explain the observed range in the fluid inclusion stable isotope values, with variable initial water contents between 0.5 and 3 μl/g. We have further added a graphical explanation of the fluid inclusion entrapment under evaporative conditions (Fig. R1).

Analytical handling is a potential process that could lead to leaking of fluid inclusions during heating. As outlined in the response to referee #1, we regard this as a less likely option. We have always monitored the water mixing ratio in the gas stream through the isotope analyser after insertion of the sample pieces. We never observed anomalies, such as sudden increases in the water vapour concentration (e.g., by cracking of large inclusion), nor corresponding changes in the isotopic values of the background water vapour. Samples which yielded evaporated water isotope values behaved similar to samples that yielded non-evaporated values. For additional illustration, Fig. R3 shows the time every sample was heated until it was crushed. If the evaporation and/or the measured water content would be related with the heating in the fluid inclusion line, we would expect a correlation between the heating time and the isotope values and/or the water content. As the attached plot shows, this is not the case. Please find further explanations and how we will include this aspect in the revised manuscript in the response to reviewer 1.

There is no graph of calculated temperature vs age showing the variations with time. I think that such a graph should be added to the paper. This graph could show the disequilibrium temperatures of the second group as well the "good' temperatures of the first group.

Fig. 5f (main manuscript) shows already the fluid-inclusion-based temperatures vs. age. However, due to potentially wrong inferences made by non-expert readers, we refrain from plotting the unrealistic temperatures in the revised version of the main manuscript. In addition, the large spread in temperatures of several tens of degrees inhibits to see the differences between the "good" temperatures in the 1-3°C range. Instead, in Fig. 5 in the main manuscript, we highlighted the intervals, where temperature reconstruction is not possible as "Larga cave ventilation events". For the interested reader, we will plot the calculated "good" and "bad" temperatures along with the speleothem fluid inclusion and calcite

proxy data ($\delta^{18}$O, $\delta^{13}$C and Mg/Ca values) in supplementary Figure S3 in the revised manuscript (added as Fig. R6 to this response letter).

Also, I am not completely clear how the authors calculated temperatures for the cold period samples. Did they use the water d18O value given by the intersection of the evaporation and meteoric water lines (Fig 3b)? If so, I would point out that one could run several different evaporation lines through the data. As Fig 3b shows, the evaporation line they calculated averages out quite widely disperse data.

No, as specified in L170ff the temperature of the comparably cold periods from cluster (1) were also directly calculated from the fluid inclusion water $\delta^{18}$O value and the corresponding calcite $\delta^{18}$O. We did not report temperatures for cluster (2) samples, which show evaporation effects.

**Additional references:**

Deininger, M., Fohlmeister, J., Scholz, D., and Mangini, A.: Isotope disequilibrium effects: The influence of evaporation and ventilation effects on the carbon and oxygen isotope composition of speleothems -- A model approach, Geochimica et Cosmochimica Acta, 96, 57-79, 2012.

Dreybrodt, W., and Scholz, D.: Climatic dependence of stable carbon and oxygen isotope signals recorded in speleothems: From soil water to speleothem calcite, Geochimica et Cosmochimica Acta, 75, 734-752, 10.1016/j.gca.2010.11.002, 2011.

Dreybrodt, W.: Implication to 13C, 18O, and clumped 13C18O isotope composition in DIC and calcite, Acta Carsologica, 48, 2019.

Hansen, M., Scholz, D., Froeschmann, M.-L., Schöne, B. R., and Spötl, C.: Carbon isotope exchange between gaseous CO 2 and thin solution films: Artificial cave experiments and a complete diffusion-reaction model, Geochimica et Cosmochimica Acta, 211, 28-47, 10.1016/j.gca.2017.05.005, 2017.

Majoube, M.: Fractionnement en oxygène 18 et en deutérium entre l'eau et sa vapeur, Journal de Chimie Physique, 68, 1423-1436, 1971.

Mühlinghaus, C., Scholz, D., and Mangini, A.: Modelling stalagmite growth and $\delta$13C as a function of drip interval and temperature, Geochimica et Cosmochimica Acta, 71, 2780-2790, 10.1016/j.gca.2007.03.018, 2007.

Scholz, D., Mühlinghaus, C., and Mangini, A.: Modelling $\delta$13C and $\delta$18O in the solution layer on stalagmite surfaces, Geochimica et Cosmochimica Acta, 73, 2592-2602, 10.1016/j.gca.2009.02.015, 2009.

Vogel, N., Scheidegger, Y., Brennwald, M. S., Fleitmann, D., Figura, S., Wieler, R., and Kipfer, R.: Stalagmite water content as a proxy for drip water supply in tropical and subtropical areas, Climate of the Past, 9, 1-12, 2013.

Warken, S., Vieten, R., Winter, A., Spötl, C., Miller, T., Jochum, K. P., Schröder-Ritzrau, A., Mangini, A., and Scholz, D.: Persistent link between Caribbean precipitation and Atlantic Ocean circulation during the Last Glacial revealed by a speleothem record from Puerto Rico, Paleoceanography and Paleoclimatology, 30, https://doi.org/10.1029/2020PA003944, 2020.

Weißbach, T.: Spectroscopic isotope ratio analysis on speleothem fluid inclusions-analytics and paleoclimatic case studies, Dissertation, Institute of Environmental Physics, Ruprecht-Karls-Universität Heidelberg, Heidelberg, 2020.